# Ethanol induces subcellular trafficking of the RNA-binding protein, hnRNP A1, in neuronal cells *in vitro*, but not in the peripubertal rat brain

Angela H. S. Fan*, Yoldas Yildiz*, Amanda A. Hartoun, Mikayla L. Newby, Rujuta Durwas, Yan Ngai, Sarah Flury and Toni R. Pak‡

## ABSTRACT

The RNA-binding protein, heterogeneous nuclear ribonucleoprotein A1 (hnRNP A1), plays a critical role in RNA metabolism, including splicing, stabilization, and transport. hnRNP A1 predominantly resides in the cell nucleus; however, it can be dynamically trafficked to the cytoplasm in response to cellular stressors, such as osmotic or oxidative stress. Although the cytoplasmic functions of hnRNP A1 are not well understood, our previous work demonstrated that hnRNP A1 associates with mature microRNAs in the cytoplasm, including those that are regulated by adolescent binge-pattern alcohol use. Therefore, this study tested the effects of repeated binge-pattern ethanol (EtOH) exposure during adolescence on hnRNP A1 subcellular localization in the rat brain and in neuronal-derived cell lines. Our results showed that EtOH exposure induced hnRNP A1 re-localization from the nucleus to the cytoplasm in neuronal cell lines, but not in the rat brain. Moreover, the primary end metabolite of EtOH, acetate, failed to induce hnRNP A1 re-localization in neuronal cell lines, suggesting that EtOH metabolism *in vivo* abrogated hnRNP A1 subcellular trafficking. We also observed that EtOH-induced hnRNP A1 re-localization to the cytoplasm was correlated with increased neuronal cell volume, suggesting that osmotic stress could be a cellular stressor driving subcellular trafficking in neurons. Notably, our study also revealed that there are brain region- and sex-specific differences in hnRNP A1 expression levels in the adolescent rat brain.

KEY WORDS: RNA-binding proteins, Stress, Alcohol, Ethanol, Puberty, Brain, Neuron, HnRNP A1

## INTRODUCTION

The heterogeneous nuclear ribonucleoprotein A1 (hnRNP A1) plays a critical role in all aspects of RNA metabolism, including transcriptional regulation, RNA splicing, intracellular RNA trafficking, and microRNA biogenesis (Clarke et al., 2021). hnRNP A1 is mainly localized in the cell nucleus but can shuttle from the nucleus to the cytoplasm in response to cellular stressors, such as heat shock, low-dose UV, osmotic and oxidative stress, which can then potentiate long-term neuronal dysfunction (Clarke et al., 2021). hnRNP A1 is one of the most abundant ribonucleoproteins (RNPs) with high expression in neuronal cells. As such, hnRNP A1 has recently been recognized as a major contributor of neuronal dysfunction resulting from cellular stress, as its re-localization from the nucleus to the cytoplasm can lead to aberrant RNA splicing and protein translation (Cammas et al., 2016; Guil et al., 2006; Salapa et al., 2024)

Our research has shown that neuron-specific mature microRNAs (miRs) are regulated by and interact with hnRNP A1 (Linscott et al., 2023). Specifically, siRNA-mediated knockdown of hnRNP A1 decreased levels of pri-miR-9-5p transcription in the nucleus, but increased levels of mature miR-9-5p in the cytoplasm. These data suggested that hnRNP A1 could regulate microRNA biogenesis at multiple levels in different compartments of the cell; however, the cellular conditions under which hnRNP A1 moves to the cytoplasm and associates with mature microRNAs are unknown. We have also shown that binge-pattern alcohol [i.e. ethanol (EtOH)] exposure during adolescence altered the expression of mature microRNAs in the hippocampal and hypothalamic brain regions – changes that persisted into adulthood (Asimes et al., 2019, 2017; Prins et al., 2014; Barney et al., 2022; Wolstenholme et al., 2017). Mature microRNAs reside primarily in the cytoplasm to regulate protein translation and are crucial for neuronal development, maintenance, and hippocampal neurogenesis throughout life (Rashidi et al., 2023). Together, these data raised the possibility that EtOH is a cellular stressor that induces hnRNP A1 trafficking from the nucleus to the cytoplasm, thereby enabling interactions with mature microRNAs.

Ethanol is a well-documented cellular stressor; yet, whether it is sufficient to induce subcellular trafficking of hnRNP A1 from the nucleus to the cytoplasm has not been tested. EtOH-induced neurological deficits can be correlated with damage in brain regions like the hippocampus and hypothalamus, which play vital roles in learning, memory, and mood regulation (Matthews and Morrow, 2000). Moreover, EtOH exposure during key neurological stages of development, such as embryonic development and adolescence, hinders adult neurogenesis, impairs learning and memory, and decreases long-term information retention (Crews et al., 2016). In the brain, EtOH is metabolized to acetaldehyde, leading to reactive oxygen species (ROS) overproduction, further exacerbating oxidative stress (Yan and Zhao, 2020). EtOH consumption contributes to increased oxidative stress by lowering levels of antioxidants, which typically neutralize ROS (Bhat et al., 2015; Tsermpini et al., 2022). Therefore, the primary objective of this study was to determine whether EtOH, acting as an oxidative and/or osmotic stressor, alters the subcellular localization of hnRNP A1 in neurons. We used hippocampal- and hypothalamic-derived neuronal cell lines, along with a well-established rat model of

Loyola University Chicago Stritch School of Medicine, Department of Cell and Molecular Physiology, Maywood, IL 60153, USA.
*These authors contributed equally to this work

‡Author for correspondence (tpak@luc.edu)

T.R.P., 0000-0002-9685-9754

repeated binge-pattern EtOH exposure, to test the hypothesis that EtOH alters hnRNP A1 localization in neurons. Our results showed that EtOH significantly increased hnRNP A1 trafficking from the nucleus to the cytoplasm in neuronal cells, but not in the brains of adolescent rats subjected to repeated binge-pattern EtOH alcohol exposure. Therefore, we next tested the effects of acetate, a terminal metabolite of EtOH, in neuronal cells and demonstrated that the effects of acetate mimicked those of EtOH in the rat model. These results suggested that *in vivo* EtOH metabolism mitigated EtOH-induced hnRNP A1 subcellular trafficking in the rat brain. Our data also revealed that there is a significant sex difference in hnRNP A1 abundance in the peripubertal rat brain.

## RESULTS
### Acute EtOH treatment increased extranuclear localization of hnRNP A1 in neuronal cells
To test if EtOH induces intracellular trafficking of hnRNP A1 from the nucleus to the cytoplasm, we first examined hnRNP A1 expression and localization following EtOH exposure in neuronal cells derived from the hippocampus (HT-22) and the hypothalamus (IVB). Cells were treated with 50 mM EtOH for 1 or 2 h. Under homeostatic conditions, hnRNP A1 is mainly localized to the nucleus, allowing for potential facilitation of mRNA transport and influencing mRNA translation. Our results confirmed that approximately 95% of hnRNP A1 was in the nucleus in the absence of EtOH in both IVB and HT-22 cells (Figs 1 and 2).

Immunofluorescence results revealed that EtOH exposure significantly increased hnRNP A1 abundance outside the nucleus in HT-22 cells (Fig. 1) and IVB cells (Fig. 2). Notably, in HT-22 cells, we observed an approximate 30% decrease in nuclear:total hnRNP A1 after 1 h and 40% after 2 h (Fig. 1D). Concomitantly, total hnRNP A1 area increased by 50 $\mu m^2$ after 1 h of EtOH treatment and 100 $\mu m^2$ after 2 h (Fig. 1C), indicative of potentially increased cell size and osmotic stress. We observed an increase in hnRNP A1 total area by 150 $\mu m^2$ and 100 $\mu m^2$ after 1-h and 2-h EtOH exposure, respectively, but saw no changes in the nuclear area of hnRNP A1 (Fig. 1B). Similarly, IVB cells showed a comparable pattern, with a 50% reduction in nuclear:total hnRNP A1 after 1 h and a 60% reduction after 2 h of EtOH exposure (Fig. 2D). Importantly, the EtOH-induced changes in subcellular localization were not due to EtOH-induced changes in total hnRNP A1 mRNA or protein levels (Fig. S1).

### EtOH treatment did not alter subcellular localization of hnRNP A1 in the hippocampus and hypothalamus of peri-pubertal rats
Next, we explored the effects of repeated binge pattern EtOH exposure on hnRNP A1 subcellular localization during the critical neurodevelopment stage of adolescence/puberty. The expression levels of hnRNP A1 protein in the brain during pubertal development have yet to be reported, nor has there been any reports of sex-specific expression in the brain. The results from our neuronal cells *in vitro* demonstrated that hnRNP A1 shifted from the nucleus to the cytoplasm after 1 and 2 h of EtOH exposure. To test if EtOH would induce a similar shift *in vivo*, we used a peri-pubertal rat model of repeated binge alcohol exposure, previously established in our laboratory, to measure hnRNP A1 protein levels in the hippocampus and hypothalamus (Asimes et al., 2019; Prins et al., 2014; Przybycien-Szymanska et al., 2014). Our results showed that hnRNP A1 was predominantly localized in the nucleus in control animals that received water alone in both brain regions (Fig. 3A-D and Fig. 4A-D), and this expression pattern was not altered following EtOH treatment in either males or females (Fig. 3A-D and Fig. 4A-D). However, we observed significantly higher levels of hnRNP A1

abundance in male compared with female rats in the hippocampus (Fig. 3E). Conversely, overall hypothalamic hnRNP A1 abundance was higher in females (Fig. 4E) than in males, demonstrating sex- and brain region-specific differences in hnRNP A1 protein levels.

### Acetate treatment did not alter hnRNP A1 localization
Our immunofluorescence findings demonstrated that hnRNP A1 localized to the cytoplasm after 1 and 2 h of EtOH exposure (Figs 1 and 2); however, this was not replicated *in vivo* in our peri-pubertal rat models (Figs 3 and 4). Intracellularly, EtOH is metabolized to acetaldehyde and then acetate, which then gets exported into the bloodstream. To determine if the differences between our cell models and our animal paradigm were due to differences in EtOH metabolism, we tested if acetate induced cellular trafficking of hnRNP A1 from the nucleus to the cytoplasm in our neuronal cells. Our results showed that, unlike EtOH, acetate did not induce hnRNP A1 re-localization from the nucleus to the cytoplasm (Fig. 5A and Fig. 6A). An acetate dose-dependent treatment was performed on HT-22 and IVB cells, and we observed that acetate concentrations between 5 mM and 30 mM did not alter hnRNP A1 localization at both 1 and 2 h in HT-22 cells (Fig. S2). However, there was a modest, yet statistically significant, difference in IVB cells at the highest dose (Fig. 6). Immunofluorescence quantification demonstrated that hnRNP A1 protein was approximately 95% nuclear in HT-22 and IVB cells following acetate treatment (Fig. 5B and Fig. 6B). Furthermore, cytoplasmic hnRNP A1 levels remained unchanged with increasing acetate concentrations during both 1- and 2-h exposure periods (Fig. 5C and Fig. 6C). Unlike EtOH treatment, acetate did not induce a substantive increase in cell size (Fig. 5D and Fig. 6D).

## DISCUSSION
Our study revealed three novel findings that add to our understanding of hnRNP A1 subcellular trafficking in the brain. First, we demonstrate the first evidence that hnRNP A1 levels in the hippocampus and hypothalamus are sex specific. We observed overall higher abundances of hnRNP A1 protein in female hypothalamus compared to male hypothalamus, but the opposite was observed in the hippocampus. Recent studies have implicated hnRNP A1 as a potential contributor to the progression of neurodegenerative diseases, and our results raise the intriguing possibility that hnRNP A1 might be an underlying driver of sex biases in neurological diseases. Second, our data revealed that EtOH induced subcellular trafficking of hnRNP A1 in neuronal cells and caused a massive shift to extranuclear areas. These data are consistent with other studies demonstrating that subcellular localization of hnRNP A1 is altered in response to cellular stress and suggest that EtOH could work through cellular stress pathways that trigger re-localization. Finally, our data also revealed that the *in vitro* effects of EtOH on hnRNP A1 localization could not be recapitulated in an animal model. These data likely reflect differences in EtOH metabolism and underscore the complexity of *in vivo* systems that protect against toxic insults, thereby preserving cellular homeostasis.

This study, to our knowledge, is the first to report sex- and brain region-specific variations in EtOH-induced subcellular hnRNP A1 localization. Biological sex differences can largely be attributed to variations in circulating sex steroid hormones, including androgens and estrogens. These hormones rapidly increase at the time of puberty and remain consistently high throughout reproductive life. Moreover, there is evidence that these hormonal differences are also at the root of sex biases in many diseases. The animals in our study were euthanized at 42 days of age, which, in the rat, is coincident with the peripubertal stage of rising sex steroid hormones (Bell, 2018). However, further studies would need to be done at different ages, and following the

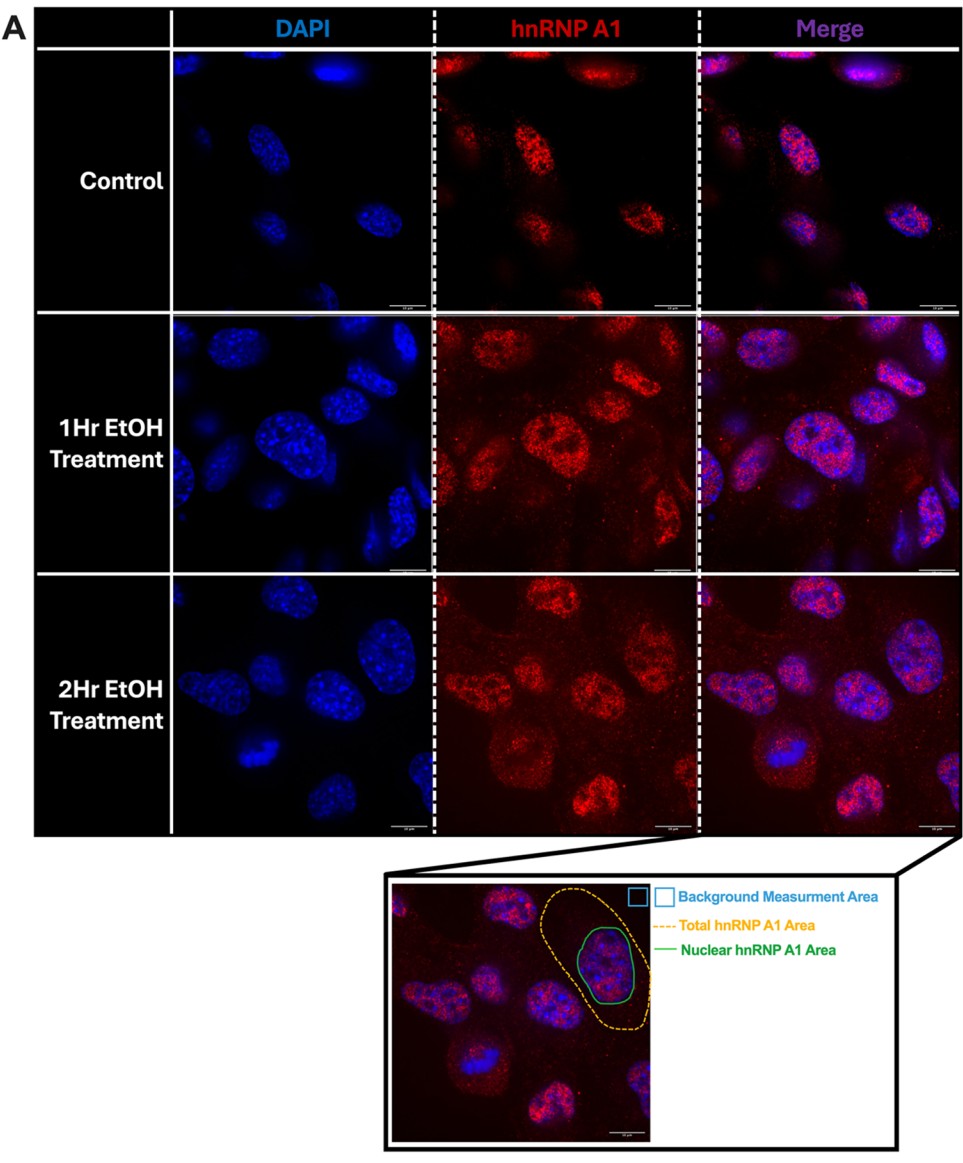

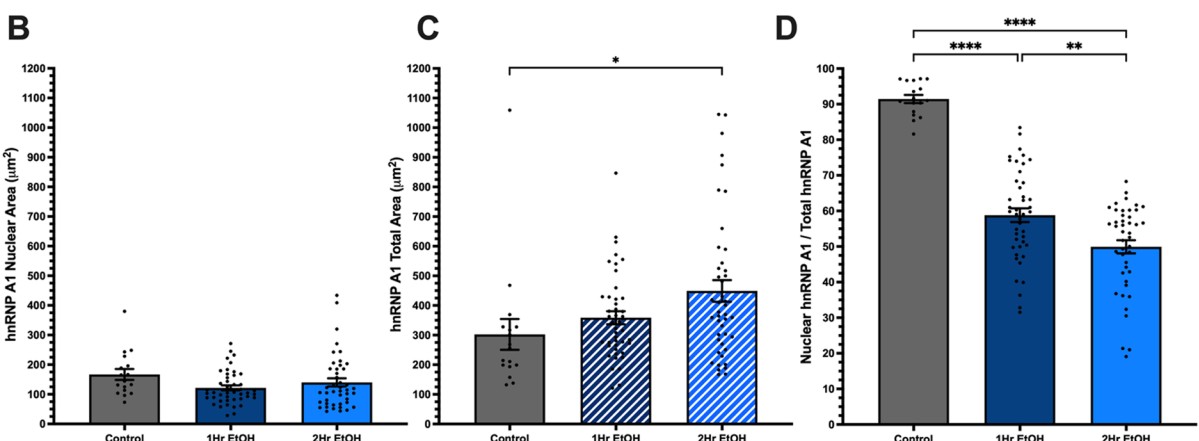

**Fig. 1. Ethanol (EtOH)-induced hnRNP A1 extranuclear localization in HT-22 cells.** (A) Representative images of HT-22 cells treated for 1 or 2 h with 50 mM EtOH. Blue, DAPI; red, hnRNP A1. (B-D) Quantification of nuclear (B) and total (C) hnRNP A1 expression and nuclear:total hnRNP A1 ratio (D). Inset shows representative quantification area as described in the Materials and Methods. Data are depicted as mean±s.e.m. and analyzed by one-way ANOVA with GraphPad Prism software. $N$=3 (6) biological (technical within each biological) replicates. *$P<0.05$, **$P<0.01$, ****$P<0.0001$. Scale bars: 10 μm.

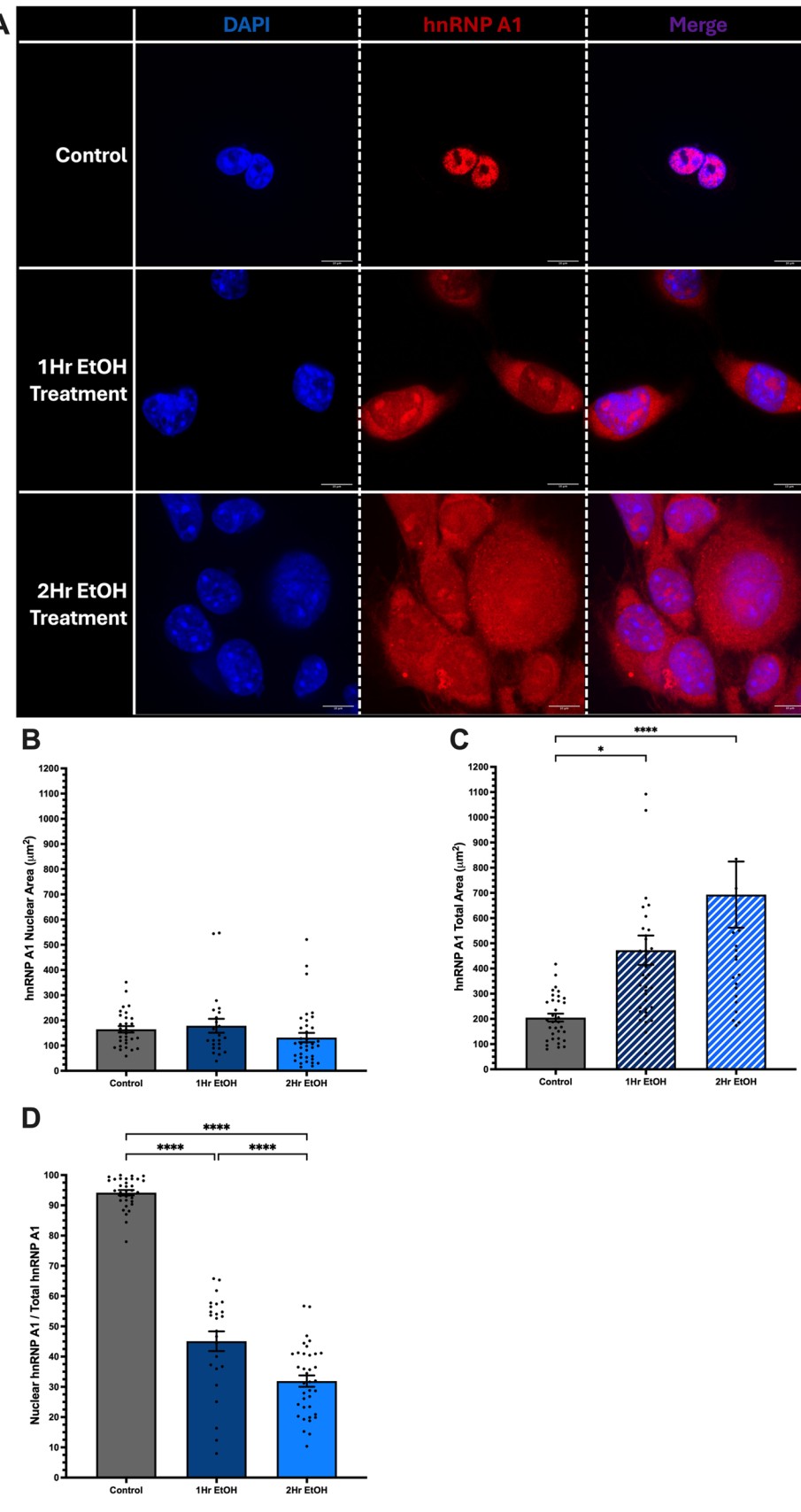

**Fig. 2. Ethanol (EtOH)-induced hnRNP A1 extranuclear localization in IVB cells.** (A) Representative images of fixed IVB cells treated for 1 or 2 h with 50 mM EtOH. Blue, DAPI; red, hnRNP A1. (B-D) Quantification of nuclear (B) and total (C) hnRNP A1 expression and nuclear:total hnRNP A1 ratio (D). Data are depicted as mean±s.e.m. and analyzed by one-way ANOVA with GraphPad Prism software. N=3 (6) biological (technical within each biological) replicates. *P<0.05, ****P<0.0001. Scale bars: 10 μm.

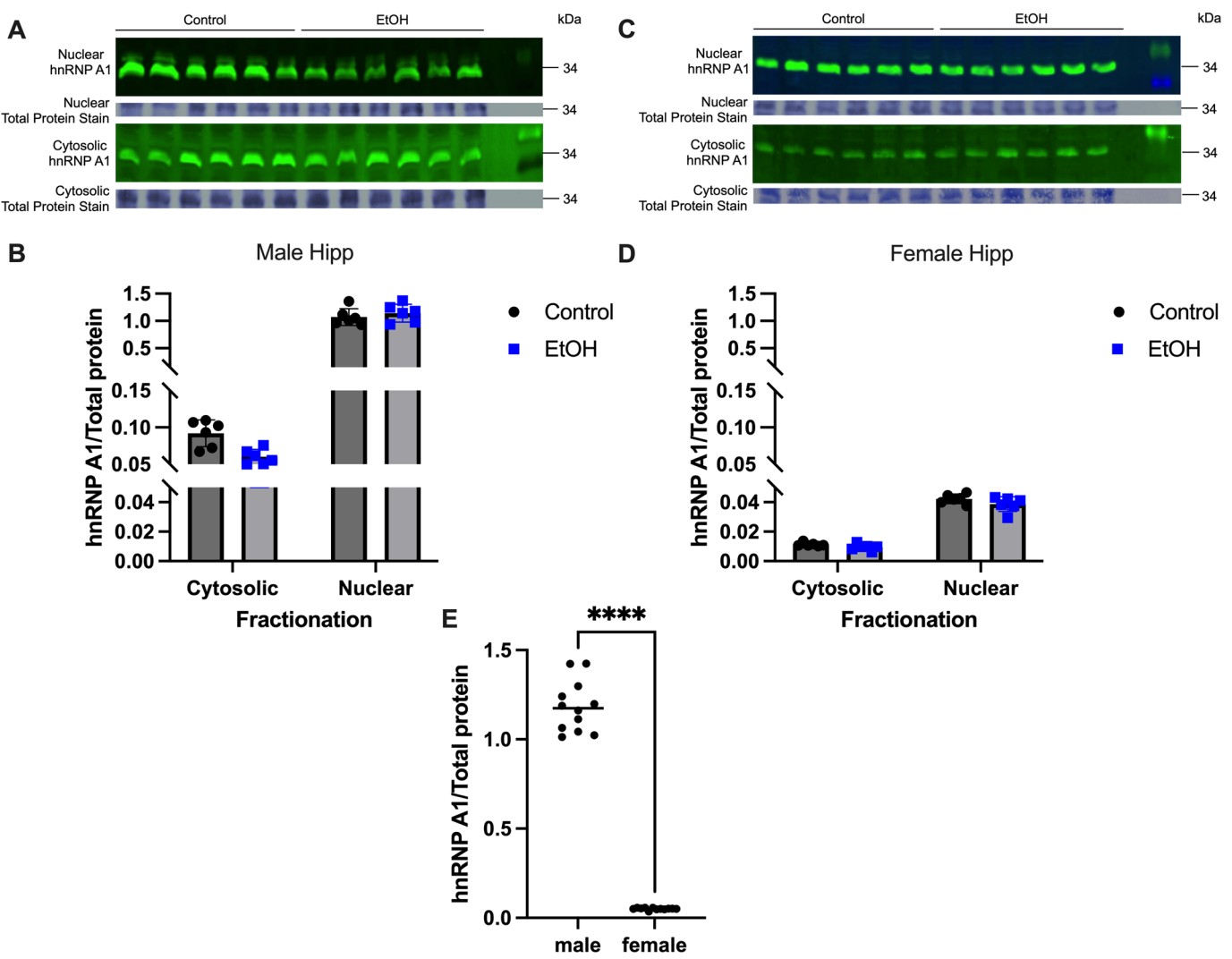

**Fig. 3. Repeated binge-EtOH exposure did not induce hnRNP A1 localization in male and female rat hippocampus.** (A-D) hnRNP A1 levels were analyzed by Western blotting in male (A,B) and female (C,D) hippocampus, quantified using AzureSpot Pro software. Data were normalized to total protein levels in each respective cellular fraction. Data are depicted as mean±s.e.m. and analyzed by two-factor ANOVA with GraphPad Prism software. No significant differences were detected between water- and EtOH-treated animals. (E) Total hnRNP A1 (cytosolic+nuclear) levels for males and females. Water- and EtOH-treated animals were analyzed together by two-tailed $t$-test. ****$P<0.0001$.

manipulation of sex steroid hormones, to conclude if hormones are the basis for our observation that hnRNP A1 expression in these brain regions is different in females compared to males. Nevertheless, estrogens have well-documented neuroprotective effects (Green and Simpkins, 2000), so these findings might have significant biological implications and warrant further study.

EtOH is a well-documented cellular stressor, as it induces both oxidative and osmotic stress (Tsermpini et al., 2022; Huang et al., 2009; Gemma et al., 2006; Hernandez et al., 2016). Previously, we reported that EtOH regulated the expression of mature microRNAs and that these microRNAs were associated with the RNA-binding protein, hnRNP A1 (Linscott et al., 2023; Kim et al., 2020; Guil and Caceres, 2007), and others have shown that cellular stress can induce hnRNP A1 trafficking to the cytoplasm (Clarke et al., 2021; Guil et al., 2006).

Therefore, we reasoned that EtOH, acting as a stressor, would shift hnRNP A1 to the cytoplasm, where it could be spatially positioned to interact with and regulate mature microRNAs. Indeed, in this study, we showed that EtOH induced a statistically significant re-localization of hnRNP A1 from the nucleus to the cytoplasm in neuronal cells. We also observed a significant increase in cell size (i.e. swelling) in response to

EtOH treatment, suggesting that EtOH might have disrupted osmotic homeostasis, and this was sufficient to induce hnRNP A1 re-localization. This interpretation is consistent with van der Houven van Oordt et al. (2000), who showed that sorbitol-induced osmotic stress caused cytoplasmic accumulation of hnRNP A1 in NIH-3T3 (mouse embryonic fibroblast) and COS (green monkey kidney) cells. Moreover, they showed that this effect was dependent on p38 kinase activation, which EtOH has also been shown to activate in HT-22 cells (Ku et al., 2006) and human monocytes (Norkina et al., 2008). Therefore, the effects of EtOH on hnRNP A1 subcellular trafficking are likely mediated by a convergence of intracellular osmotic and oxidative stress signaling pathways.

Several factors may have contributed to the discrepancies observed between our neuronal cell model and animal experiments, but differences in EtOH metabolism are likely the primary factor. EtOH metabolism primarily occurs in the liver through two key steps: the oxidation of EtOH to acetaldehyde, followed by the oxidation of acetaldehyde to acetate (Tsermpini et al., 2022). EtOH in rats undergoes a first-pass metabolism in the stomach and liver (Levitt et al., 1994), accounting for up to 90% of total alcohol metabolism

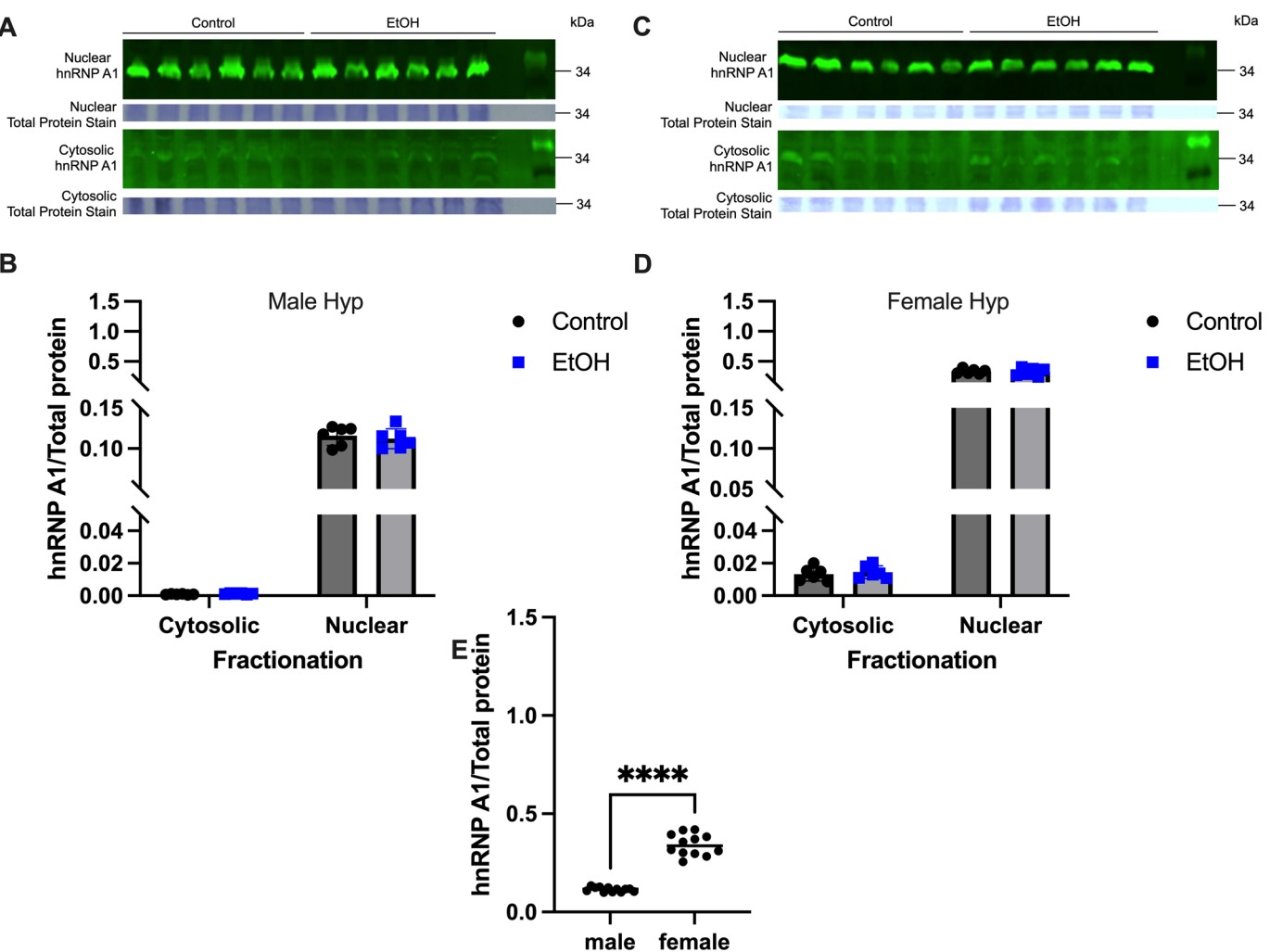

**Fig. 4. Repeated binge-EtOH exposure did not induce hnRNP A1 localization in male and female rat hypothalamus.** (A-D) (A-D) hnRNP A1 levels were analyzed by Western blotting in male (A,B) and female (C,D) hypothalamus, quantified using AzureSpot Pro software. Data were normalized to total protein levels in each respective cellular fraction. Data are depicted as mean±s.e.m. and analyzed by two-factor ANOVA with GraphPad Prism software. No significant differences were detected between water- and EtOH-treated animals. (E) Total hnRNP A1 (cytosolic+nuclear) levels for males and females. Water- and EtOH-treated animals were analyzed together by two-tailed $t$-test. ****$P<0.0001$.

(Cederbaum, 2012; Holford, 1987). Consequently, approximately ~10% of EtOH that crosses the blood-brain barrier (if not lost from urine, sweat, or breath) reaches the brain, potentially insufficient to induce a level of osmotic dysregulation required to stimulate hnRNP A1 re-localization. In contrast, our neuronal cell lines bypassed first-pass metabolism, resulting in EtOH concentrations sufficient to trigger cell swelling and potentially activate osmotic/oxidative stress downstream signaling pathways. This interpretation is supported by our results showing that acetate treatment in the neuronal cells did not induce subcellular trafficking of hnRNP A1. Moreover, acetate has been shown to reduce ROS, thereby having an antioxidant effect, which, *in vivo* would compensate for the excess ROS generated by EtOH in the cells. Taken together, these results strongly suggest that differences in the kinetics of metabolism, and therefore the actual concentration of EtOH received by the brain cells, are the most likely explanation for the discrepancy in our results. These results also emphasize that caution must be used when interpreting results obtained with EtOH in neuronal cells and extrapolating that to what occurs in the whole brain of intact animals.

An outstanding question is how hnRNP A1 accumulates in the cytoplasm. Two potential mechanisms are that cellular stress actively

shuttles hnRNP A1 out of the nucleus or that it leads to rapid degradation of hnRNP A1 within the nucleus, followed by the prevention of newly translated hnRNP A1 proteins from re-entering, and evidence supports both. For instance, hyperphosphorylation may alter hnRNP A1 subcellular distribution by reducing its nuclear import rate (Allemand et al., 2005; Hock et al., 2018), which would result in cytoplasmic accumulation. Conversely, poly(ADP-ribosyl)ation (PARylation) modulates hnRNP A1 localization by regulating its nucleocytoplasmic transport, where PAR-binding via the PAR-binding motif drives hnRNP A1 out of the nucleus to recruit it to stress granule formation (Duan et al., 2019). The degradation and turnover rates of hnRNP A1 in the nucleus are not well understood, nor is there good agreement about the cellular function of cytoplasmic stress granule formation in response to certain cellular stressors. Either way, cytoplasmic accumulation of hnRNP A1, whether sequestered in stress granules or not, disrupts normal RNA metabolism and contributes to potentially cytotoxic protein aggregates. Future work will be important to discern whether some types of cellular stress, such as EtOH and its metabolites, prevent nuclear import and others facilitate nuclear export, especially if considering hnRNP A1 as a therapeutic target for disease.

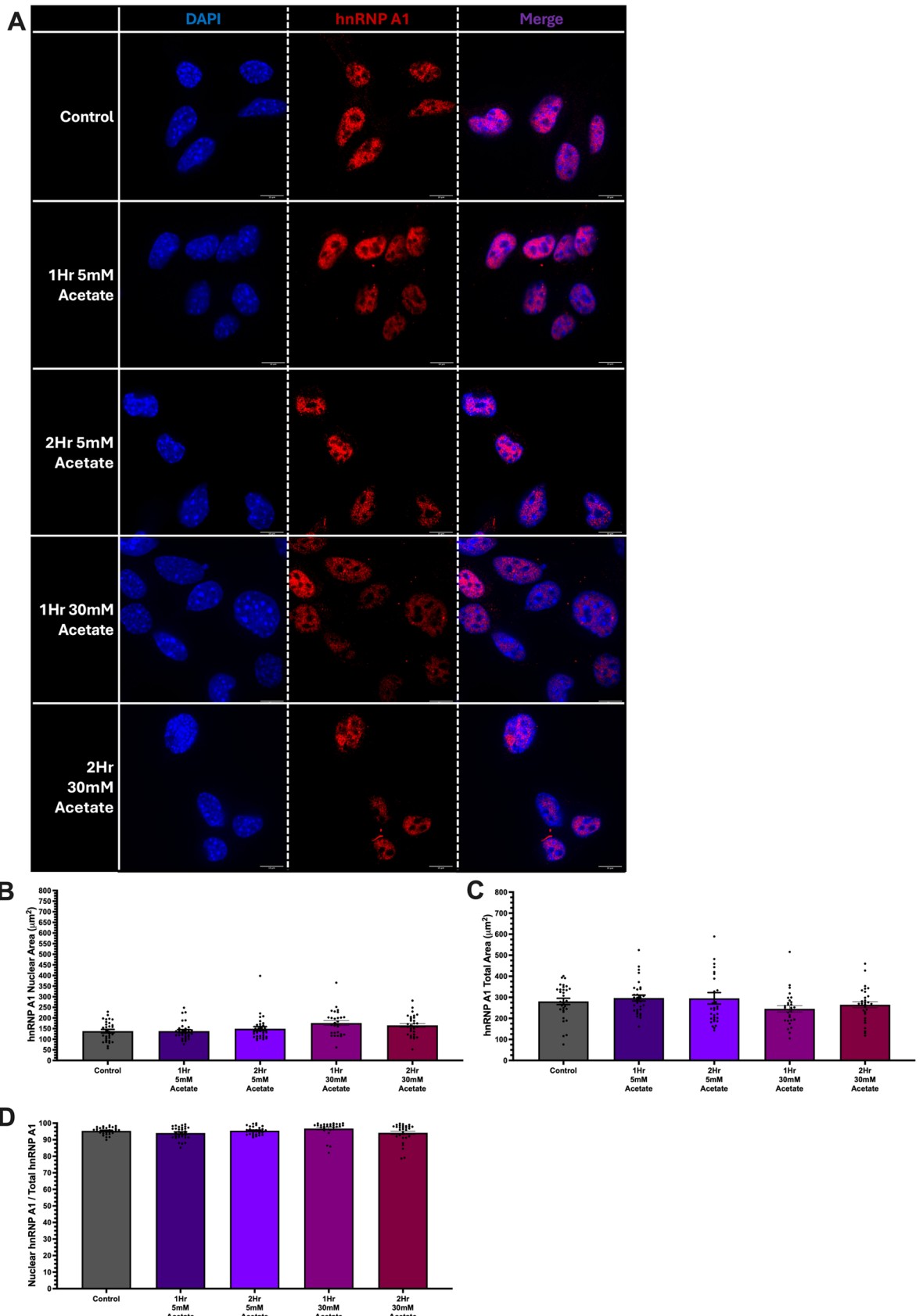

**Fig. 5. Acetate did not induce hnRNP A1 extranuclear localization in HT-22 cells.** (A) Representative images of fixed HT-22 cells treated for 1 or 2 h with 5 mM or 30 mM sodium acetate. Blue, DAPI; red, hnRNP A1. Scale bars: 10 µm. (B,C) Nuclear (B) and total (C) hnRNP A1 were quantified using ImageJ software. (D) Nuclear hnRNP A1 compared to total area. Data are depicted as mean±s.e.m. $N$=3 (6) biological (technical within each biological) replicates. No statistically significant differences were observed.

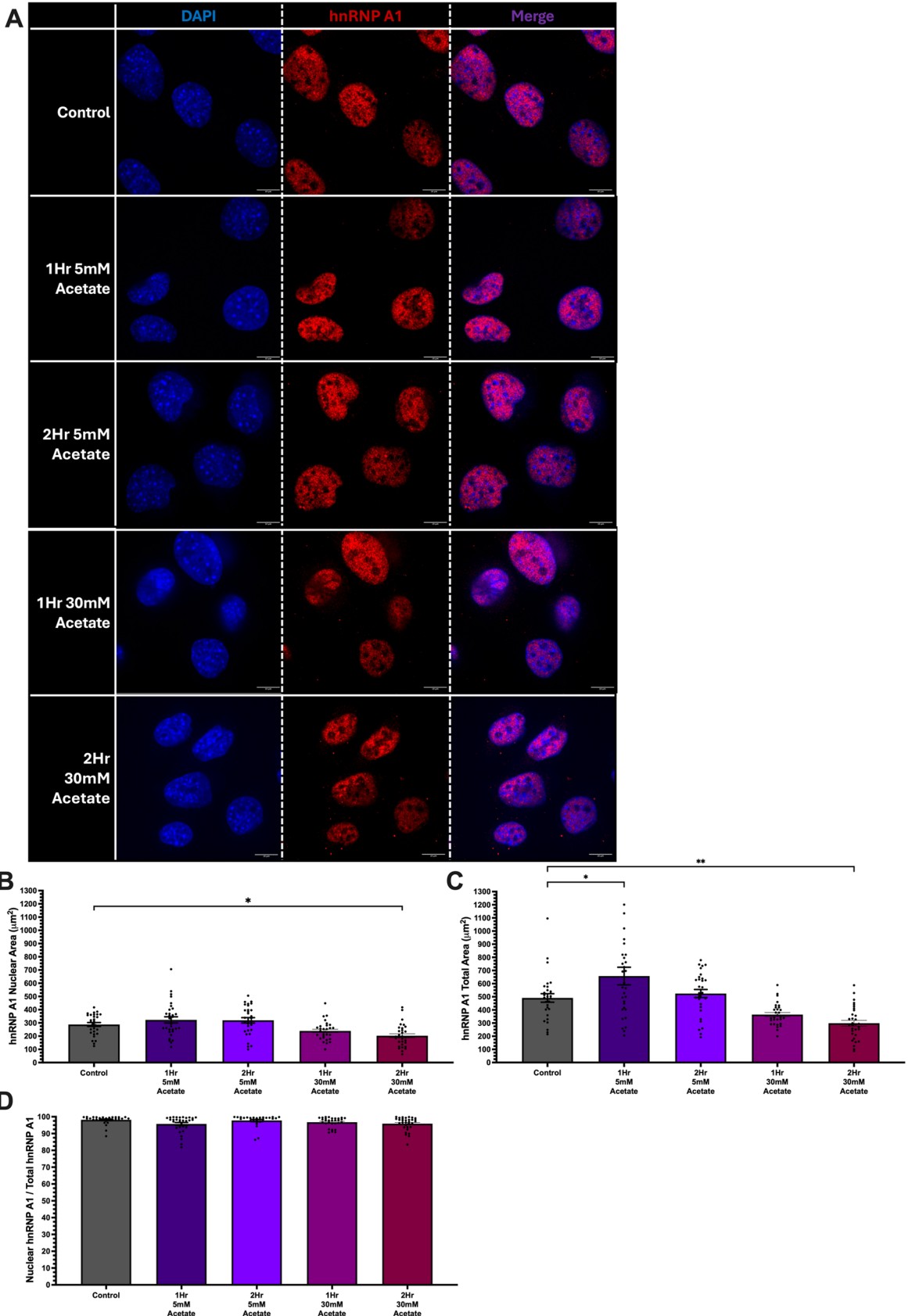

**Fig. 6. Acetate did not induce hnRNP A1 extranuclear localization in IVB cells.** Representative images of fixed IVB cells treated for 1 or 2 h with 5 mM or 30 mM sodium acetate. Blue, DAPI; red, hnRNP A1. Scale bars: 10 μm. (B,C) Nuclear (B) and total (C) hnRNP A1 were quantified using ImageJ software. (D) Nuclear hnRNP A1 compared to total area. Data are depicted as mean±s.e.m. *N*=3 (6) biological (technical within each biological) replicates. No statistically significant differences were observed. *$P<0.01$, **$P<0.001$.

Adolescent binge drinking has been a growing national health concern over the past several decades. Mounting evidence has shown that this pattern of alcohol consumption during the critical period of pubertal brain development results in long-term neurological deficits. Our laboratory and others have worked to pinpoint the molecular targets mediating these detrimental consequences of alcohol consumption. The broad and vital cellular functions regulated by hnRNP A1 position it as a potential critical node for the molecular underpinnings of EtOH's deleterious effects. Moreover, its cytoplasmic accumulation results in insoluble protein aggregates, allowing it to potentially dysregulate microRNAs, facilitate stress granule formation, or impede ER protein folding. Together, this cascade of failing cellular machinery can ultimately lead to cell death and, in the case of neurons, disrupted synaptic connections. Here, our study demonstrates that EtOH has the potential to induce hnRNP A1 re-localization in neuronal cells, but whether this occurs in the context of an intact physiological systems requires further investigation.

## MATERIALS AND METHODS

### Cell culture

*In vitro* studies were conducted using a rat neuronal cell line (IVB), derived from the paraventricular nucleus of the rat hypothalamus (generously gifted from Dr John Kaskow, University of Cincinnati, Cincinnati, OH, USA), and a mouse neuronal cell line (HT-22) derived from the hippocampus (generously gifted from Dr Dave Schubert, Salk Institute, San Diego, CA, USA). IVB cells were grown and maintained normal media consisting of Dulbecco's modified Eagle medium (DMEM), 10% heat-inactivated fetal bovine serum (FBS) (BenchMark™ GeminiBio, 100-106) and Plasmocin™ prophylactic (25 mg/500 ml; Invivogen, ant-mpp). HT-22 cells were grown and maintained in normal medium consisting of DMEM, 10% heat-inactivated FBS, 1% non-essential amino acids (Sigma-Aldrich, M7145-100ML), and Plasmocin™ prophylactic (25 mg/500 ml; Invivogen, ant-mpp). Cells were tested and verified mycoplasma-free prior to all experiments (InvivoGen, Mycotrip, rep-mys-50). All experiments were performed on cells within ten passages, and each experiment was repeated using a minimum of three different passages (for biological, $N=3$) and six technical replicates within each biological $N$.

### Cell treatment: EtOH and acetate

IVB and HT-22 cells were seeded at 250,000 cells per well in a six-well plate and grown to 80-90% confluency before treatment. Cells were treated with medium containing either 50 mM EtOH, 100 mM EtOH, or normal growth medium as a negative control for 1 or 2 h at 37°C. Acetate treatment was applied at varying concentrations (5 mM, 10 mM, 15 mM, 20 mM, 25 mM and 30 mM). The concentration of acetate did not affect hnRNP A1 localization (Fig. S2); therefore, 1 and 2 h of 5 mM and 30 mM acetate treatments are shown as representative images in Figs 5 and 6. Cells were then processed for hnRNP A1 immunofluorescence, reverse transcription-quantitative polymerase chain reaction (RT-qPCR), and Western blot analyses as described below.

### Immunofluorescence

Following EtOH and acetate treatments, cells were washed in PBS and fixed in 4% paraformaldehyde for 20 min. Cells were then permeabilized with 1% Triton-X in Tris-buffered saline (TBS-T) for 10 min and washed with TBS three times for 5 min. Cells were then incubated in blocking buffer (10% normal goat serum in 1% TBS-T) for 1 h at room temperature, followed by incubation of primary mouse hnRNP A1 monoclonal antibody (Santa Cruz Biotechnology, sc-32301) at 1:250 in blocking buffer overnight at 4°C. Cells were then washed with TBS 3×5 min at room temperature. Following washing, slides were incubated with secondary goat anti-mouse Alexa Fluor 647 (Thermo Fisher Scientific, A32728) at 1:1000 in blocking buffer for 2 h at room temperature followed by TBS wash 3×5 min. A drop of prolonged diamond antifade mountant with DAPI (Thermo Fisher Scientific, P36962) was applied to the coverslips and cured on microscope slides at room temperature for 1 h. Slides were then sealed with nail polish and stored at −20°C until imaging.

### Imaging and quantification

Immunostained fixed cells were imaged on a Marianas Imaging System (Intelligent Imaging Innovations) consisting of an Axio Observer 7 (Zeiss) microscope base, a CSU-W1 SoRa Spinning Disk (Yokogawa), a laser launch that includes 405, 488, and 637 nm lasers and Slidebook software (Intelligent Imaging Innovations). Images were collected using a Plan-Apochromat 63×/1.46 NA oil TIRF objective, using the SoRa 1× and 2.8× multiplier on a Prime 95B sCMOS (Photometrics) camera. Quantification of hnRNP A1 was performed by drawing two regions in each cell, one that outlined the nucleus (based on the DAPI channel) and one that contained all the hnRNP A1 signal in each cell. We then summed the intensity of the hnRNP A1 signal in both regions, subtracted the background signal, and divided them to calculate the ratio as follows:

$$ratio = \frac{I_{nuclearA1} - bg*A_{nuclear}}{I_{totalA1} - bg*A_{total}}.$$

Quantification of nuclear:total fluorescence of hnRNP A1 was calculated using ImageJ software.

### RNA isolation and cDNA synthesis

After treatment, medium was removed, and cells were washed with sterile 1× PBS and collected in 500 µl TRI Reagent (Zymo Research, R2050-1-200) for RNA isolation. Samples were briefly sonicated and processed according to the manufacturer's instructions. The concentration and purity of RNA was determined using a Nanodrop spectrophotometer (Thermo Fisher Scientific). Total RNA (600 ng) was used to synthesize cDNA following the SSIV cDNA synthesis protocol (Thermo Fisher Scientific, 18091050). The cDNA was then stored at −20°C.

### RT-qPCR

Relative levels of hnRNP A1 were measured using RT-qPCR, with hypoxanthine phosphoribosyl transferase 1 (*Hprt1*) used as a housekeeping gene. Our previous studies have shown that EtOH treatment does not alter HPRT expression (Prins et al., 2014). In brief, a master mix was made using nuclease-free water, the forward and reverse primers for hnRNP A1 or *Hprt1*, and SYBR Green Master Mix (Thermo Fisher Scientific). Samples were analyzed by quantitative PCR using a QuantStudio 3 Real-Time Quantitative PCR system (Applied Biosystems) under the following conditions: (1) 95°C for 10 min, (2) 95°C for 30 s, (3) 60°C for 30 s, and (4) 72°C for 30 s, in addition to melting curve analysis. Transcripts were quantified as log2 fold change relative to vehicle-treated control using the ΔΔCt method (Livak and Schmittgen, 2001).

### Animals (*N*=24)

Male (*N*=12) and female (*N*=12) Wistar rats were purchased from Charles River Laboratory (Wilmington, MA, USA) at postnatal day (PND) 25 and were allowed to acclimate for 5 days. Animals were pair-housed within the same treatment group. Food and water were available *ad libitum*, and animals were kept on a 12:12 light/dark cycle, with lights on at 07:00 and handling/treatment beginning at 10:00. Animal procedures were approved by the Loyola University Medical Center Institutional Animal Care and Use Committee (IACUC; #2022023).

All measures were taken to minimize pain and suffering. Animals would be excluded if (1) removed prior to end of the study for health reasons as detailed in the approved animal protocol, (2) they did not receive correct treatment dose as determined by the experimenter, (3) or data were flagged as outliers by Prism statistical software. These criteria were set prior to the experiment. No animals or data points were removed. Animals were pair-housed based on similar weights and randomly assigned to 'water' or 'EtOH' groups based on cage placement on a rack.

### Animal handling period

At PND 30, animals were handled for 5 min 1×/day for 7 days to control for nonspecific handling stress. On PND 35, animals were introduced to the oral gavage tool by placing it gently in their mouths and allowing them to nibble on the tube. On PND 36, the animals were administered a 'dry' oral gavage in which the syringe did not contain any liquid.

## EtOH treatment paradigm

Beginning at PND 37, which is defined as peri-puberty in the rat (Ketelslegers et al., 1978), animals were exposed to a repeated binge-pattern alcohol paradigm (Fig. S3). This 8-day paradigm has been used previously by our laboratory and others and is designed to mimic the reported drinking patterns of adolescents (Przybycien-Szymanska et al., 2014, 2010; Lauing et al., 2008). This pattern of alcohol consumption raised the blood alcohol concentration (BAC) to 259 mg/dl in males and 279 mg/dl in females (Fig. S4). We have previously shown that this 1×/day short-term treatment did not alter body weight or normal growth patterns (Przybycien-Szymanska et al., 2010; Lauing et al., 2008). Animals were given food-grade alcohol (Everclear™, Luxco, St. Louis, MO, USA) diluted in tap water at a dose of 3.2 g/kg body weight (20% v/v solution) or an equal volume of vehicle (water) via oral gavage. Treatment was administered at the same time each day (1×/day for 3 days, followed by 1×/day tap water for 2 days, and then 1×/day for 3 days alcohol [total $N$=12 (6/sex)]. Control groups received equal volumes of tap water 1×/day for 8 days [total $N$=12 (6/sex)]. Sample size was based on prior studies (Przybycien-Szymanska et al., 2010). Animals were humanely euthanized 60 min after the last treatment on PND 44 by exposure to inhaled isoflurane until unconsciousness, followed by rapid decapitation.

## BAC assay

Trunk blood samples were collected into heparinized tubes, centrifuged at 3000 RPM for 10 min at 4°C, and the plasma was stored at −20°C. Blood alcohol levels were determined by measuring the change in absorbance at 340 nm following enzymatic oxidation of EtOH to acetylaldehyde (Point Scientific Alcohol Reagent Kit). Assay range is 0-400 mg/dl, and intra- and inter-assay CV=6.4% and 7.9%, respectively.

## Western blotting

### Sample preparation: cellular protein extraction

Adherent cells were harvested using 0.05% trypsin and collected in complete medium. Cells were pelleted by centrifugation at 1000 RPM for 5 min. Cell lysates were stored at −80°C until total protein quantification was determined using a colorimetric bicinchoninic acid (BCA) assay according to the manufacturer's instructions.

### Sample preparation: tissue protein extraction and fractionation

Whole hypothalamus or hippocampus was freshly dissected, transferred to a microcentrifuge tube containing 1 ml PBS, and centrifuged at $500\,g$ for 5 min. The supernatant was removed, leaving a dry tissue pellet. Tissue homogenization was performed using 600 µl ice-cold CER I supplemented with 100× protease inhibitor. 33 µl CER II and 300 µl NER were added into each tube and vortexed for 5 s, incubated on ice for 1 min, then the tubes were centrifuged for 5 min at 16,000 $g$ at 4°C. The supernatant (cytoplasmic extract) was immediately transferred to a clean pre-chilled tube on wet ice. The pellet (insoluble fraction that contains the nuclei) was suspended in ice-cold NER, vortexed for 15 s, and placed on ice, continuing vortexing for 15 s every 10 min, for a total of 40 min. The tubes were centrifuged at 16,000 $g$ for 10 min at 4°C. The supernatant (nuclear extract) was immediately transferred to a clean pre-chilled tube on wet ice until storage in −80°C or total protein quantification, performed using BCA assay according to the manufacturer's instructions. The nuclear fractions were briefly sonicated before proceeding to protein quantification assays.

### Western blot analysis

30 µg of protein per sample was loaded onto 12% gel and then transferred onto a PVD membrane (Thermo Fisher Scientific, 88518) for 1 h at 100 V. The amount of protein per sample needed for hnRNP A1 antibody was calculated using a calibration curve (see data in doi:10.6084/m9.figshare.28148216). To normalize target signals, a total protein stain was performed according to the manufacturer's protocol (LI-COR Revert™ 700 Total Protein Stain Kit). The PVD membrane was blocked with blocking buffer (LI-COR Intercept® Part No. 927-60001) and TBS-T for 1 h and then incubated with mouse hnRNP A1 primary antibody (Santa Cruz Biotechnology, sc-32301, 1:100) overnight at 4°C. Following incubation, the PVD membrane was washed three times for 10 min in 10 ml TBS-Tween (Tris Base solution containing 0.1% Tween 20), incubated in donkey anti-mouse secondary antibody (LI-COR IRDye®

800CW P/N: 926-32212) for 1 h at room temperature, and washed three times for 10 min in TBS. The PVD membrane was then imaged using Azure Sapphire FL Biomolecular Imager (RGBNIR, Dublin, CA, USA) according to the manufacturer's directions. hnRNP A1 protein levels were quantified and normalized to total protein using AzureSpot Pro software (Azure Biosystems, Dublin, CA, USA).

## Statistical analysis

Statistical analyses were performed using PRISM software (v10.4.0; GraphPad Software, San Diego, CA, USA). Experimenters were unaware of treatments during data analysis. Data were analyzed by one-way and two-way ANOVA unless otherwise noted in figure captions. Data are displayed as mean±s.e.m., and statistical significance was determined at $P<0.05$.

## Rigor and reproducibility

The following measures are embedded within our experimental designs to ensure that they adhere to the highest standards for rigorous scientific methods: (1) experimenters will be unaware of all animal/cell treatments for endpoint experiments and data analysis; (2) animal numbers were calculated using statistical power analyses that ensured the minimum number in order to achieve statistically and biologically significant results; (3) we use an age-appropriate animal model that we have already characterized with gonad histology and hormone profiles and physiologically relevant EtOH dose/delivery method; and (4) we use established hippocampal and hypothalamic neuronal cell lines to ensure rigorous design and reproducibility for mechanistic studies that are not feasible in whole animals. Data are collected, analyzed, and stored in an electronic notebook (LabArchives) that is openly shared with members of the investigative team. To ensure transparency, data will be freely disseminated using database depositories according to National Institutes of Health (NIH) guidelines, and data reporting adhere to Animal Research: Reporting of *In Vivo* Experiments (ARRIVE) guidelines.

## Competing interests

The authors declare no competing or financial interests.

## Author contributions

Conceptualization: A.H.S.F., Y.Y., T.R.P.; Formal analysis: A.H.S.F., Y.Y., A.A.H., M.L.N., R.D., Y.N., S.F., T.R.P.; Funding acquisition: T.R.P.; Investigation: A.H.S.F., Y.Y., A.A.H., M.L.N., Y.N., S.F.; Methodology: A.H.S.F., Y.Y., S.F., T.R.P.; Project administration: S.F., T.R.P.; Resources: T.R.P.; Supervision: T.R.P.; Validation: S.F., T.R.P.; Visualization: A.H.S.F., Y.Y.; Writing – original draft: A.H.S.F., Y.Y., T.R.P.; Writing – review & editing: A.H.S.F., Y.Y., A.A.H., M.L.N., R.D., Y.N., S.F., T.R.P.

## Funding

This research was supported by the National Institute on Alcohol Abuse and Alcoholism (R01AA021517 to T.R.P.). Open Access funding provided by Loyola University Chicago Stritch School of Medicine. Deposited in PMC for immediate release.

## Data and resource availability

The data that support the findings of this study are available at doi:10.6084/m9. figshare.28148216. All other relevant data can be found within the article and its supplementary information.

## First person

This article has an associated First Person interview with the joint first authors of the paper.

## Peer review history

The peer review history is available online at https://journals.biologists.com/bio/lookup/doi/10.1242/bio.062010.reviewer-comments.pdf

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
