## [Peer Review File · Biology Open]

Ethanol induces subcellular trafficking of the RNA-binding protein, hnRNP A1, in neuronal cells in vitro, but not in the peripubertal rat brain

Angela H.S. Fan, Yoldas Yildiz, Amanda A. Hartoun Mikayla L. Newby, Rujuta Durwas, Yan Ngai, Sarah Flury and Toni R Pak
DOI: 10.1242/bio.062010

Editor: Sandhya Koushika

Review timeline

Original submission:	1 April 2025
Editorial decision:	9 April 2025
First revision received:	10 June 2025
Accepted:	12 June 2025

Original submission

First decision letter

MS ID#: bio.062010

MS Title: Ethanol induces subcellular trafficking of the RNA-binding protein, hnRNP A1, in neuronal cells

Authors: Angela H.S. Fan; Yoldas Yildiz; Amanda A. Hartoun; Mikayla L. Newby; Rujuta Durwas; Yan Ngai; Sarah Flury; Toni R Pak

Dear Dr Pak,

I have now reached a decision on the above manuscript.

The reviewer reports are shown at the bottom of this email or can be accessed, together with a copy of this decision letter, by going to:

.

As you will see, the reviewers gave favourable reports, but raised some critical points that will require amendments to your manuscript. I hope that you will be able to carry these out, because we would like to be able to accept your paper.

I request you to assess major comment 3 of reviewer 1 carefully especially as pertains to bar graphs in Fig 1B and 2B. Reviewer 2 comment 5: the claims can be modified to account for the concern raised without doing additional experiments.

Reviewer 1

Comments for the author

Here, Fan, Yildiz et al attempt to test a hypothesis in two parts: that ethanol exposure induces cytoplasmic trafficking of hnRNP A1: (a) in vitro, and (b) in vivo in a rat binge ethanol exposure paradigm, as a potential explanation for previously observed miRNA expression changes. In

particular, the authors and others previously showed that binge ethanol exposure induces miRNA changes in the rat brain (Prins et al, 2014; Asimes et al., 2019; Barney et al, 2022), while they separately showed that hnRNP A1 knockdown in IVB neuronal cells in vitro led to changes in levels of its interaction partner miR-9-5p (Linscott et al., 2023; Kim et al, 2020). The authors show here that ethanol exposure induced increased trafficking of hnRNP A1 to the cytoplasm in vitro, but not in vivo using their binge ethanol exposure rat model. Instead, they report interesting pre-existing sex-differences in hnRNP A1 expression levels in the rat brain. Aspects of this work will require technical clarifications and modifications to strengthen the results.

Major comments

1. Regarding the choice of ethanol concentration in vitro:
 - a. Page 3, Lines 10-11: It will be useful for the authors to discuss why 50 mM Ethanol was chosen for the in vitro assay, perhaps with respect to physiological relevance. It is noted that both 50 mM and 100 mM conditions were used for Fig S4 (and as stated in the Methods), whereas 50 mM only was used for Figs 1 and 2. Perhaps stating the rationale for this will be useful.
 - b. Page 5, Lines 20-21: the authors astutely discuss that 90% of ethanol is lost to first pass metabolism, and only ~10% or less will reach the brain. In this context, perhaps a dose-dependent ethanol assay would have been the more appropriate assay in vitro, to discover the threshold required for hnRNP A1 cytoplasmic localisation, and to reconcile this with the 10% physiological ethanol concentration that reaches the brain, and the lack of hnRNP A1 cytoplasmic localisation in vivo. In the absence of this data, the utility of an in vitro dosage experiment can be discussed instead.
2. A key technical aspect of Figures 1 and 2 that I struggle to understand even after reading the Methods is how the IF cell boundaries were determined (Page 7, Line 11: "Total cell fluorescence was measured by outlining individual cells in ImageJ "). The most accurate way is the use of a separate Phase Contrast imaging or a separate plasma membrane marker channel (etc.) to robustly segment individual cells. It is ill-advised to rely on the hnRNP A1 channel, as it is absent in the cytoplasm in control treatment, and may not be localised evenly in the cytoplasm in ethanol treatment unless this has been ascertained appropriately previously. It will therefore be important to clarify how this was done in the Methods, and having these phase contrast/cell boundary/traced outline images in Fig 1A and 2A as an extra column will be helpful.
3. On scrutinising the images in Fig 1A vs 2A, it may appear that there is clearer and higher levels of efflux of hnRNP A1 to the cytoplasm in IVB cells compared to HT-22 cells. However, the bar plot for Fig 1B and 2B suggest that both cell lines had identical cytoplasmic efflux observations. On very close inspection, it seems that the bar plot in Fig 2B may be a potential duplicate of 1B as the scatter of data points appear identical and super-imposable. It seems there was an honest error in figure assembly for Fig 2B, as the barplots for respective bars in 1B and C add up correctly to 100%, while the bars in barplots 2B and 2C do not (they sum to below 100% by eye for the 1hr and 2hr EtOH bars). Once corrected, perhaps this observation can be more accurately discussed.
4. Supplementary Figure 4 is an important one that shows no change in total protein levels of hnRNP A1 in the in vitro assay. This figure should be renamed as "Supplementary Figure 1" in keeping with the flow of the manuscript. In addition, it will be useful to show the representative Western Blots for these bar plots (for Fig A and C). It will also be interesting to note if the authors had any comment on the differential relative hnRNP A1 protein levels between the two cell lines (particularly if they are from a different sex, and as they're from different parts of the brain).
5. For figures 3A, 3C, 4A and 4C, it will be helpful to include the respective "total protein stain" images so that the reader can more confidently appreciate the normalised quantifications in the rest of the figure. This can be a simple crop of the total protein stain around 34 kDa (similar to how manuscripts represent Ponceau stains as total protein loading controls). This will aid in interpretation of Fig 3A where slightly decreased nuclear hnRNP A1 protein bands are visible, yet no difference is noted upon normalisation in Fig 3B, which probably indicates lower total protein loading for those lanes.
6. As a corollary, the authors observed a highly interesting sex-specific difference in total hnRNP A1 protein levels between male and female brain regions, which they claim to be

"statistically significant" (page 3, Line 49). However this requires the reader to look at separate bar plots (Fig 3/4A vs 2/4B) to make this judgment with no statistical test of p-values shown for this particular male vs female comparison. It will therefore be important to add an extra panel E to both Fig 3 and 4 to compare the total protein levels of hnRNP A1 directly between male vs females in the respective brain regions with the deployment of the appropriate statistical test, and an explanation of the p-values.

7. To further corroborate the sex-differences observed for hnRNP A1 levels, would the authors have access to qPCR data of Hnrnpa1 mRNA levels to establish whether the difference is a transcriptional one? Or perhaps a reanalysis of a pre-existing RNA-seq dataset for male versus female (rat or mouse) hippocampal or hypothalamic brain regions at the time point assayed?

8. While it is unfortunate that binge ethanol exposure did not lead to hnRNP A1 cytoplasmic trafficking in vivo, it remains of interest that this effect was seen in vitro which the authors discuss from Page 5, Line 39. Two further outstanding questions come to mind which may be discussed briefly in the discussion:

a. Does the in vitro ethanol experiment impact miRNA levels, in particular miR-9-5p levels? It may be useful to consider how best to test if hnRNP A1's localisation to the cytoplasm may influence this.

b. As hnRNP A1 levels and localisation remain unchanged in the ethanol binge exposure model in vivo, what alternative explanations remain for the miRNA changes observed in Prins et al, 2014; Asimes et al., 2019 and Barney et al, 2022?

9. The authors may need to tone down their claims "indicating that osmotic stress was the main cellular stressor driving subcellular trafficking in neurons" in the abstract. Perhaps "indicating" can be replaced by "suggesting", particularly as osmotic stress was not directly tested in this work, with knowledge of ethanol's multifarious roles in stress induction, which the authors acknowledge well in the discussion.

Minor comments

1. Page 3, Line 10: It will be important to mention at the onset that IVB and HT-22 cell lines are mouse-derived, and to also specify the sex (XX or XY) of these cell lines. This will aid the reader to reconcile certain findings, particularly as the in vivo paradigm is of a different species (rat), and that sex-specific differences were being discussed with respect to rat neuronal hnRNP A1.

2. Supplementary Fig 2 shows the scheme of the ethanol binge exposure paradigm. However, it includes 3 experimental groups: "repeat binge", "acute binge" and "water control". In this work, only "repeat binge" and "water control" are investigated. It will be important to revise this scheme to remove the "acute binge" paradigm that was not used in this work.

Reviewer 2

Comments for the author

The manuscript by Fan et al. titled "Ethanol induces subcellular trafficking of the RNA-binding protein, hnRNP A1, in neuronal cells" investigates how ethanol and acetate supplementation affect the subcellular localisation of the hnRNP A1 protein in neuronal cells. The authors conducted substantive experiments, both in cell culture and in rat models, to assess how hnRNP A1 localisation responds to ethanol. Overall, the manuscript presents interesting new data and underscores significant differences between cell culture and animal experiments in response to ethanol stress. The authors present data showing that ethanol treatment in neuronal cell culture dramatically changes the localisation of hnRNP A1 protein, and this change correlates with the dramatic change in cell size likely due to osmotic stress. Whereas acetate, which forms during ethanol metabolism, doesn't affect hnRNP A1 localisation.

Furthermore, the authors didn't observe any change in the localisation of hnRNP A1 in rat models of binge drinking, where the animals were given ethanol over the course of time. This part of the manuscript is important to highlight how cell culture and animal model experiments can differ in

their outcome. However, some of the experiments in this section using subcellular fractionation of tissue proteins lack sufficient controls.

The authors should consider addressing the following points before publication;

1. The title does not accurately reflect the observations of the manuscript. There is no data in the manuscript to suggest hnRNP A1 "induces subcellular trafficking". Similar wording is problematic throughout the manuscript but actually discussed in the discussion section correctly. In addition, the title should reflect that the observation is limited to neuronal cells in culture. This is because the lack of similar observations in the animal model used is an important part of the manuscript. Perhaps "Ethanol induces cytoplasmic localisation of hnRNP A1 in neuronal cells in culture but not in animal models of binge drinking"
2. In Figures 1 and 2, the authors show that treating cells with 50 mM EtOH for 1 or 2 hours alters hnRNP A1 localisation. Can the authors explain why this specific concentration was chosen and why they did not conduct a dose-response analysis for the observed phenotype? Additionally, can the authors include an example of how they measured the areas, perhaps by providing supplemental data of the same figures with nuclear and cytoplasmic regions highlighted as they did in ImageJ? Finally, I could not find the statistical tests used for Figures 1 and 2.
3. On page 3, relating to figures 3 and 4, the authors should clarify in the text which figure shows which sample, i.e., hippocampus vs. hypothalamus.
4. Figures 3 and 4 lack sufficient information. For instance, the authors do not present any data regarding the quality of their hippocampus or hypothalamus isolations, nor do they demonstrate the effectiveness of their nuclear and cytoplasmic fractionations. This could have been easily accomplished by blotting for tissue-enriched and nuclear or cytoplasmic-enriched proteins.
5. The assertion that hnRNP A1 levels vary based on animal sex is ambiguous because the effectiveness of subcellular fractionation across experiments remains unknown. If the authors include controls to demonstrate the levels of nuclear and cytoplasmic factors and normalise hnRNP A1 levels to these, then this assertion would be better substantiated.

Reviewer's Responses to Questions

Experimental quality

Does each figure have the proper controls?

If 'No', please indicate reasons in Comments for Author box below.

Reviewer #1:

- Yes

Reviewer #2:

- No

Were the data analyzed using appropriate statistical tests?

If 'No', please indicate reasons in Comments for Author box below.

Reviewer #1:

- No

Reviewer #2:

- No

Reproducibility

Were experiments performed using adequate number of biological replicates?
If 'No', please indicate reasons in Comments for Author box below.

Reviewer #1:

- No

Reviewer #2:

- Yes

Does the methods section provide sufficient detail to permit reproducibility?
If 'No', please indicate reasons in Comments for Author box below.

Reviewer #1:

- Yes

Reviewer #2:

- Yes

Completeness

Are the manuscript's conclusions supported by the data?
If 'No', please indicate reasons in Comments for Author box below.

Reviewer #1:

- Yes

Reviewer #2:

- No

Scholarship

Do the authors cite and discuss the merits of data that would argue for and against their conclusion?
If 'No', please indicate reasons in Comments for Author box below.

Reviewer #1:

- Yes

Reviewer #2:

- Yes

Does the manuscript title & abstract accurately reflect the contents of the manuscript, without hyperbole?

If 'No', please indicate reasons in Comments for Author box below.

Reviewer #1:

- No

Reviewer #2:

- No

First revision

Author response to reviewers' comments

We thank the reviewers for their thorough consideration of our manuscript and helpful comments. Below, we have first addressed comments raised by both reviewers followed by a point-by-point response for unique comments from each. We are confident that the changes we made to the manuscript has improved its clarity and sufficiently addressed all concerns.

Comments from both reviewers

1) Explanation of immunofluorescent quantification:

The reviewers are correct that our described method of quantification was confusing. Specifically, the word “cytoplasmic” was misleading because as pointed out, we did not have a separate channel to show the clear shape of the cell. Our main goal with these data was to show that more hnRNP A1 is found outside of the nucleus in our EtOH treatments compared to control. To quantify this, we drew two regions in each cell, one that outlined the nucleus (based on the DAPI channel) and one that contained all the hnRNP A1 signal in a given cell. We then summed the intensity of the hnRNP A1 signal in both of these regions, subtracted the background signal, and divided them to calculate the ratio:

$$ratio = \frac{I_{nuclearA1} - bg * A_{nuclear}}{I_{totalA1} - bg * A_{total}}$$

While this does not necessarily represent the entire cytoplasmic compartment, it does accurately convey our main point that more hnRNP A1 is found outside the nucleus in our EtOH treated groups. We did not quantify signal from cells that were clearly touching and/or on top of other cells. In response to the reviewers comments we have updated the details of our approach in the methods section. We have also removed the word cytoplasmic and relabeled the axis as “Nuclear hnRNP A1 / Total hnRNP A1” and reduced the number of plots to simply show the ratio, and the areas of the two regions. Finally, we have also incorporated the reviewer’s excellent suggestion to show example regions in the representative images (see new Figure 1).

2) Quantification of sex differences:

As suggested by the reviewers, we reanalyzed our data to directly compare sex differences. Because there was no statistical difference between cytosolic and nuclear fractions, we combined both fractions to determine total hnRNP A1 levels in each brain region. Moreover, our statistical analysis did not show any difference between water and EtOH-treated animals. Therefore, to increase the statistical power of our test, we combined water and EtOH treated animals for this comparison. The resulting analysis revealed a highly statistically difference by sex which is now shown in Figs. 3 and 4 (panel E).

3) Justification for *in vitro* dose:

The 50 mM dose was chosen based on calculations that were consistent with the BAC levels in our *in vivo* studies (~0.23%). There has been some disagreement in the field about whether the brain “sees” the amount of EtOH that is often measured *in vivo* as BAC. For this reason, we chose to keep the levels equivalent. Our results suggested that perhaps the brain wasn’t exposed to that concentration of EtOH, but instead a metabolite, which is why we followed up with the acetate experiments. We also used a 100 mM dose to ensure that EtOH had no adverse effects on hnRNP A1 mRNA or protein, but this dose is considered supraphysiological.

Unique comments - Reviewer 1:

Comment 1. As stated above, the 50 mM dose was equivalent to the dose used *in vitro*. We also used 100 mM dose *in vitro* to determine if EtOH altered hnRNPA1 total protein and decided to use a supraphysiologically high dose to rule out any effects even at very high doses.

Comment 2 - addressed above

Comment 3. Thank you for pointing out the error with Fig. 1B and 2B. It was indeed a mistake when compiling the figures and we have now corrected it with the correct graph for Fig. 2B.

Comment 4. We have changed the order of supplementary figures as suggested by the reviewer.

Comment 5. As suggested, we have included a representative total protein panel along with our representative western blot for each brain region.

Comment 6. Sex differences concern explained above.

Comment 7. Unfortunately, we do not have any samples remaining from these animals to perform a qPCR for hnRNPA1 mRNA.

Comment 8a. This is an excellent question and one we plan to follow up on in future studies. Unfortunately, we used all the available brain tissue samples from these animals, and this requires a new cohort for experiments.

8b. Thank you for this comment. We have performed some preliminary studies using mass spectrometry to take an unbiased approach to identify EtOH-induced protein changes specific to the miRNA biogenesis pathway. These data are being analyzed and validated and will be the subject of a different manuscript.

Comment 9. We agree with the reviewer that we cannot conclude that EtOH induced osmotic stress without directly measuring using an osmometer. Our speculation was based on the fact that EtOH treatment caused the cells to “swell”. Moreover, IVB cells are derived from the paraventricular nucleus of the hypothalamus, a known region that regulates plasma osmolality. Nevertheless, we have rewritten the discussion and modified our speculation regarding osmotic stress as suggested.

Unique Comments - Reviewer #2

Comment 1. We have changed the title to more accurately reflect our findings.

Comment 2. These concerns are addressed above, as it was noted by both reviewers.

Comment 3. We have noted the regions in figures 3 and 4 as suggested.

Comment 4. We quantified hnRNP A1 in different cellular compartments based on total protein within each compartment. The Western blots were quantified based on total protein in each cellular compartment (nucleus and cytoplasm), and not to specific protein markers. We chose this method over normalizing against specific nuclear and cytoplasmic markers for two reasons: first, normalizing hnRNPA1 to a specific protein marker only provides information on how much hnRNPA1 there is relative to that specific protein, which is a different question than what we

were asking. Instead, we wanted to determine how much hnRNPA1 was present relative to all the proteins in each cellular compartment (i.e. cytoplasm v. nucleus). This method has been systematically evaluated and was determined to be a more accurate method for quantification of Western blots, especially when using subcellular fractionation (see Pillai-Kastoori et al., 2020, Analytical Biochemistry). Second, we could not find literature directly comparing how sex and/or EtOH affects the abundance of these “common” markers (e.g. Lamin A/C or HSP60), which could skew the results if we used those markers to normalize for hnRNPA1. Unfortunately, as a result of choosing this method, and due to limited protein sample, we did not have enough left to perform western blots for specific markers in these animals.

We are confident that even if a minor contamination between fractions occurred, it would not have substantively changed the results from our *in vivo* studies. First, all samples were fractionated simultaneously so any carryover would have also occurred in our control (not EtOH-treated) samples. Second, the amount of nuclear hnRNP A1 was greater than 95% in all our *in vivo* samples, which is consistent with literature reports of where hnRNP A1 is typically localized in cells and also consistent with our immunocytochemistry data from *in vitro* control-treated neuronal cells (Figs. 1, 2, 5, 6). Finally, we have validated the efficiency of our fractionation procedure in previous studies and are confident that there is little to no carryover between fractions.

Comment 5. See response above to all reviewers.

Second decision letter

MS ID#: bio.062010R1

MS Title: Ethanol induces subcellular trafficking of the RNA-binding protein, hnRNP A1, in neuronal cells *in vitro*, but not in the peripubertal rat brain.

Authors: Angela H.S. Fan; Yoldas Yildiz; Amanda A. Hartoun; Mikayla L. Newby; Rujuta Durwas; Yan Ngai; Sarah Flury; Toni R Pak

Dear Dr Pak,

I am happy to tell you that your manuscript has been accepted for publication in Biology Open, pending our standard publication integrity checks. It was accepted on 12 Jun 2025.